# Learning shape correspondence with anisotropic convolutional neural networks

**Davide Boscaini**[1], **Jonathan Masci**[1], **Emanuele Rodolà**[1], **Michael Bronstein**[1,2,3]
[1]USI Lugano, Switzerland     [2]Tel Aviv University, Israel     [3]Intel, Israel
`name.surname@usi.ch`

## Abstract

Convolutional neural networks have achieved extraordinary results in many computer vision and pattern recognition applications; however, their adoption in the computer graphics and geometry processing communities is limited due to the non-Euclidean structure of their data. In this paper, we propose Anisotropic Convolutional Neural Network (ACNN), a generalization of classical CNNs to non-Euclidean domains, where classical convolutions are replaced by projections over a set of oriented anisotropic diffusion kernels. We use ACNNs to effectively learn intrinsic dense correspondences between deformable shapes, a fundamental problem in geometry processing, arising in a wide variety of applications. We tested ACNNs performance in challenging settings, achieving state-of-the-art results on recent correspondence benchmarks.

## 1 Introduction

In geometry processing, computer graphics, and vision, finding intrinsic correspondence between 3D shapes affected by different transformations is one of the fundamental problems with a wide spectrum of applications ranging from texture mapping to animation [25]. Of particular interest is the setting in which the shapes are allowed to deform non-rigidly. Traditional hand-crafted correspondence approaches are divided into two main categories: *point-wise correspondence methods* [17], which establish the matching between (a subset of) the points on two or more shapes by minimizing metric distortion, and *soft correspondence methods* [23], which establish a correspondence among functions defined over the shapes, rather than the vertices themselves. Recently, the emergence of 3D sensing technology has brought the need to deal with acquisition artifacts, such as missing parts, geometric, and topological noise, as well as matching 3D shapes in different representations, such as meshes and point clouds. With new and broader classes of artifacts, comes the need of learning from data invariance that is otherwise impossible to model axiomatically.

In the past years, we have witnessed the emergence of *learning-based approaches* for 3D shape analysis. The first attempts were aimed at learning local shape descriptors [15, 5, 27], and shape correspondence [20]. The dramatic success of deep learning (in particular, convolutional neural networks [8, 14]) in computer vision [13] has led to a recent keen interest in the geometry processing and graphics communities to apply such methodologies to geometric problems [16, 24, 28, 4, 26].

***Extrinsic* deep learning.** Many machine learning techniques successfully working on images were tried "as is" on 3D geometric data, represented for this purpose in some way "digestible" by standard frameworks. Su *et al.* [24] used CNNs applied to range images obtained from multiple views of 3D objects for retrieval and classification tasks. Wei *et al.* [26] used view-based representation to find correspondence between non-rigid shapes. Wu *et al.* [28] used volumetric CNNs applied to rasterized volumetric representation of 3D shapes. The main drawback of such approaches is their treatment of geometric data as Euclidean structures. Such representations are not intrinsic, and vary

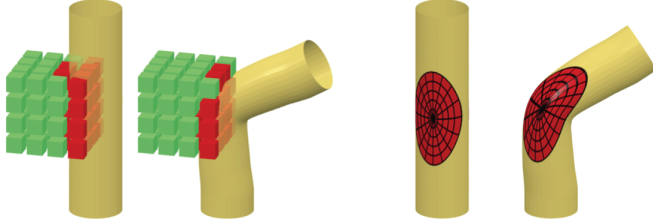

Figure 1: Illustration of the difference between extrinsic (left) and intrinsic (right) deep learning methods on geometric data. Intrinsic methods work on the manifold rather than its Euclidean realization and are isometry-invariant by construction.

as the result of pose or deformation of the object. For instance, in Figure 1, the filter that responds to features on a straight cylinder would not respond to a bent one. Achieving invariance to shape deformations, a common requirement in many applications, is extremely hard with the aforementioned methods and requires complex models and huge training sets due to the large number of degrees of freedom involved in describing non-rigid deformations.

***Intrinsic* deep learning**   approaches try to apply learning techniques to geometric data by generalizing the main ingredients such as convolutions to non-Euclidean domains. In an intrinsic representation, the filter is applied to some data on the surface itself, thus being invariant to deformations by construction (see Figure 1). The first intrinsic convolutional neural network architecture (Geodesic CNN) was presented in [16]. While producing impressive results on several shape correspondence and retrieval benchmarks, GCNN has a number of significant drawbacks. First, the charting procedure is limited to meshes, and second, there is no guarantee that the chart is always topologically meaningful. Another intrinsic CNN construction (Localized Spectral CNN) using an alternative charting technique based on the windowed Fourier transform [22] was proposed in [4]. This method is a generalization of a previous work [6] on spectral deep learning on graphs. One of the key advantages of LSCNN is that the same framework can be applied to different shape representations, in particular, meshes and point clouds. A drawback of this approach is its memory and computation requirements, as each window needs to be explicitly produced.

**Contributions.**   We present Anisotropic Convolutional Neural Networks (ACNN), a method for intrinsic deep learning on non-Euclidean domains. Though it is a generic framework that can be used to handle different tasks, we focus here on learning *correspondence* between shapes. Our approach is related to two previous methods for deep learning on manifolds, GCNN [16] and ADD [5]. Compared to [5], where a learned spectral filter applied to the eigenvalues of anisotropic Laplace-Beltrami operator, we use anisotropic heat kernels as spatial weighting functions allowing to extract a local intrinsic representation of a function defined on the manifold. Unlike ADD, our ACNN is a convolutional neural network architecture. Compared to GCNN, our construction of the "patch operator" is much simpler, does not depend on the injectivity radius of the manifold, and is not limited to triangular meshes. Overall, ACNN combines all the best properties of the previous approaches without inheriting their drawbacks. We show that the proposed framework outperforms GCNN, ADD, and other state-of-the-art approaches on challenging correspondence benchmarks.

## 2   Background

We model a 3D shape as a two-dimensional compact Riemannian manifold (surface) $X$. Let $T_x X$ denote the *tangent plane* at $x$, modeling the surface locally as a Euclidean space. A *Riemannian metric* is an inner product $\langle \cdot, \cdot \rangle_{T_x X} : T_x X \times T_x X \to \mathbb{R}$ on the tangent plane, depending smoothly on $x$. Quantities which are expressible entirely in terms of Riemannian metric, and therefore independent on the way the surface is embedded, are called *intrinsic*. Such quantities are invariant to isometric (metric-preserving) deformations.

Heat diffusion on manifolds is governed by the *heat equation*, which has the most general form

$$f_t(x,t) = -\mathrm{div}_X(\mathbf{D}(x)\nabla_X f(x,t)), \tag{1}$$

with appropriate boundary conditions if necessary. Here $\nabla_X$ and $\mathrm{div}_X$ denote the intrinsic gradient and divergence operators, and $f(x,t)$ is the temperature at point $x$ at time $t$. $\mathbf{D}(x)$ is the *thermal conductivity tensor* ($2 \times 2$ matrix) applied to the intrinsic gradient in the tangent plane. This formulation allows modeling heat flow that is position- and direction-dependent (*anisotropic*). Andreux *et*

*al.* [1] considered anisotropic diffusion driven by the surface curvature. Boscaini *et al.* [5], assuming that at each point $x$ the tangent vectors are expressed w.r.t. the orthogonal basis $\mathbf{v}_m, \mathbf{v}_M$ of principal curvature directions, used a thermal conductivity tensor of the form

$$\mathbf{D}_{\alpha\theta}(x) = \mathbf{R}_\theta(x) \begin{bmatrix} \alpha & \\ & 1 \end{bmatrix} \mathbf{R}_\theta^\top(x), \tag{2}$$

where the $2 \times 2$ matrix $\mathbf{R}_\theta(x)$ performs rotation of $\theta$ w.r.t. to the maximum curvature direction $\mathbf{v}_M(x)$, and $\alpha > 0$ is a parameter controlling the degree of anisotropy ($\alpha = 1$ corresponds to the classical isotropic case). We refer to the operator

$$\Delta_{\alpha\theta} f(x) = -\mathrm{div}_X(\mathbf{D}_{\alpha\theta}(x)\nabla_X f(x))$$

as the *anisotropic Laplacian*, and denote by $\{\phi_{\alpha\theta i}, \lambda_{\alpha\theta i}\}_{i\geq 0}$ its eigenfunctions and eigenvalues (computed, if applicable, with the appropriate boundary conditions) satisfying $\Delta_{\alpha\theta}\phi_{\alpha\theta i}(x) = \lambda_{\alpha\theta i}\phi_{\alpha\theta i}(x)$.

Given some initial heat distribution $f_0(x) = f(x, 0)$, the solution of heat equation (1) at time $t$ is obtained by applying the *anisotropic heat operator* $H_{\alpha\theta}^t = e^{-t\Delta_{\alpha\theta}}$ to $f_0$,

$$f(x, t) = H_{\alpha\theta}^t f_0(x) = \int_X f_0(\xi) h_{\alpha\theta t}(x, \xi)\, d\xi, \tag{3}$$

where $h_{\alpha\theta t}(x, \xi)$ is the *anisotropic heat kernel*, and the above equation can be interpreted as a non-shift-invariant version of convolution. In the spectral domain, the heat kernel is expressed as

$$h_{\alpha\theta t}(x, \xi) = \sum_{k\geq 0} e^{-t\lambda_{\alpha\theta k}} \phi_{\alpha\theta k}(x)\phi_{\alpha\theta k}(\xi). \tag{4}$$

Appealing to the signal processing intuition, the eigenvalues $\lambda$ play the role of 'frequencies', $e^{-t\lambda}$ acts as a low-pass filter (larger $t$ corresponding to longer diffusion results in a filter with a narrower pass band). This construction was used in ADD [5] to generalize the OSD approach [15] using anisotropic heat kernels (considering the diagonal $h_{\alpha\theta t}(x, x)$ and learning a set of optimal task-specific spectral filters replacing the low-pass filters $e^{-t\lambda_{\alpha\theta k}}$).

**Discretization.** In the discrete setting, the surface $X$ is sampled at $n$ points $V = \{\mathbf{x}_1, \ldots, \mathbf{x}_n\}$. The points are connected by edges $E$ and faces $F$, forming a manifold triangular mesh $(V, E, F)$. To each triangle $ijk \in F$, we attach an orthonormal reference frame $\mathbf{U}_{ijk} = (\hat{\mathbf{u}}_M, \hat{\mathbf{u}}_m, \hat{\mathbf{n}})$, where $\hat{\mathbf{n}}$ is the unit normal vector to the triangle and $\hat{\mathbf{u}}_M, \hat{\mathbf{u}}_m \in \mathbb{R}^3$ are the directions of principal curvature. The thermal conductivity tensor for the triangle $ijk$ operating on tangent vectors is expressed w.r.t. $\mathbf{U}_{ijk}$ as a $3 \times 3$ matrix $\begin{pmatrix} \alpha & \\ & 1 \\ & & 0 \end{pmatrix}$.

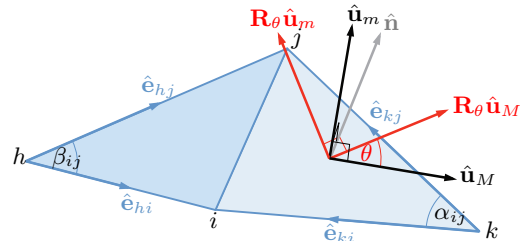

The discretization of the anisotropic Laplacian takes the form of an $n \times n$ sparse matrix $\mathbf{L} = -\mathbf{S}^{-1}\mathbf{W}$. The *mass matrix* $\mathbf{S}$ is a diagonal matrix of area elements $s_i = \frac{1}{3}\sum_{jk:ijk\in F} A_{ijk}$, where $A_{ijk}$ denotes the area of triangle $ijk$. The *stiffness matrix* $\mathbf{W}$ is composed of weights

$$w_{ij} = \begin{cases} \frac{1}{2}\left(\frac{\langle \hat{\mathbf{e}}_{kj}, \hat{\mathbf{e}}_{ki}\rangle_{\mathbf{H}_\theta}}{\sin\alpha_{ij}} + \frac{\langle \hat{\mathbf{e}}_{hj}, \hat{\mathbf{e}}_{hi}\rangle_{\mathbf{H}_\theta}}{\sin\beta_{ij}}\right) & (i, j) \in E; \\ -\sum_{k\neq i} w_{ik} & i = j; \\ 0 & \text{else}, \end{cases} \tag{5}$$

where the notation is according to the inset figure, and the shear matrix $\mathbf{H}_\theta = \mathbf{R}_\theta \mathbf{U}_{ijk}\begin{pmatrix} \alpha & \\ & 1 \\ & & 0 \end{pmatrix}\mathbf{U}_{ijk}^\top \mathbf{R}_\theta^\top$ encodes the anisotropic scaling up to an orthogonal basis change. Here $\mathbf{R}_\theta$ denotes the $3 \times 3$ rotation matrix, rotating the basis vectors $\mathbf{U}_{ijk}$ on each triangle around the normal $\hat{\mathbf{n}}$ by angle $\theta$.

# 3 Intrinsic deep learning

This paper deals with the extension of the popular convolutional neural networks (CNN) [14] to non-Euclidean domains. The key feature of CNNs is the convolutional layer, implementing the idea of "weight sharing", wherein a small set of templates (filters) is applied to different parts of the data. In image analysis applications, the input into the CNN is a function representing pixel values given on a Euclidean domain (plane); due to shift-invariance the convolution can be thought of as passing a template across the plane and recording the correlation of the template with the function at that location. One of the major problems in applying the same paradigm to non-Euclidean domains is the lack of shift-invariance, the template now has to be location-dependent.

Among the recent attempts to develop intrinsic CNNs on non-Euclidean domain [6, 4, 16], the most related to our work is GCNN [16]. The latter approach was introduced as a generalization of CNN to triangular meshes based on geodesic local patches. The core of this method is the construction of local geodesic polar coordinates using a procedure previously employed for intrinsic shape context descriptors [12]. The *patch operator* $(D(x)f)(\theta, \rho)$ in GCNN maps the values of the function $f$ around vertex $x$ into the local polar coordinates $\theta, \rho$, leading to the definition of the *geodesic convolution*

$$(f * a)(x) = \max_{\Delta\theta \in [0,2\pi)} \int a(\theta + \Delta\theta, \rho)(D(x)f)(\theta, \rho)d\rho d\theta, \tag{6}$$

which follows the idea of multiplication by template, but is defined up to arbitrary rotation $\Delta\theta \in [0, 2\pi)$ due to the ambiguity in the selection of the origin of the angular coordinate. The authors propose to take the maximum over all possible rotations of the template $a(\rho, \theta)$ to remove this ambiguity. Here, and in the following, $f$ is some feature vector that is defined on the surface (e.g. texture, geometric descriptors, etc.)

There are several drawbacks to this construction. First, the charting method relies on a fast marching-like procedure requiring a triangular mesh. While relatively insensitive to triangulation [12], it may fail if the mesh is very irregular. Second, the radius of the geodesic patches must be sufficiently small compared to the injectivity radius of the shape, otherwise the resulting patch is not guaranteed to be a topological disk. In practice, this limits the size of the patches one can safely use, or requires an adaptive radius selection mechanism.

# 4 Anisotropic convolutional neural networks

The key idea of the Anisotropic CNN presented in this paper is the construction of a patch operator using anisotropic heat kernels. We interpret heat kernels as local weighting functions and construct

$$(D_\alpha(x)f)(\theta, t) = \frac{\int_X h_{\alpha\theta t}(x, \xi)f(\xi)d\xi}{\int_X h_{\alpha\theta t}(x, \xi)d\xi}, \tag{7}$$

for some anisotropy level $\alpha > 1$. This way, the values of $f$ around point $x$ are mapped to a local system of coordinates $(\theta, t)$ that behaves like a polar system (here $t$ denotes the scale of the heat kernel and $\theta$ is its orientation). We define *intrinsic convolution* as

$$(f * a)(x) = \int a(\theta, t)(D_\alpha(x)f)(\theta, t)dt d\theta, \tag{8}$$

Note that unlike the arbitrarily oriented geodesic patches in GCNN, necessitating to take a maximum over all the template rotations (6), in our construction it is natural to use the principal curvature direction as the reference $\theta = 0$.

Such an approach has a few major advantages compared to previous intrinsic CNN models. First, being a spectral construction, our patch operator can be applied to any shape representation (like LSCNN and unlike GCNN). Second, being defined in the spatial domain, the patches and the resulting filters have a clear geometric interpretation (unlike LSCNN). Third, our construction accounts for local directional patterns (like GCNN and unlike LSCNN). Fourth, the heat kernels are always well defined independently of the injectivity radius of the manifold (unlike GCNN). We summarize the comparative advantages in Table 1.

**ACNN architecture.** Similarly to Euclidean CNNs, our ACNN consists of several layers that are applied subsequently, i.e. the output of the previous layer is used as the input into the subsequent one.

| Method | Repr. | Input | Generalizable | Filters | Context | Directional | Task |
|---|---|---|---|---|---|---|---|
| OSD [15] | Any | Geometry | Yes | Spectral | No | No | Descriptors |
| ADD [5] | Any | Geometry | Yes | Spectral | No | Yes | Any |
| RF [20] | Any | Any | Yes | Spectral | No | No | Correspondence |
| GCNN [16] | Mesh | Any | Yes | Spatial | Yes | Yes | Any |
| SCNN [6] | Any | Any | No | Spectral | Yes | No | Any |
| LSCNN [4] | Any | Any | Yes | Spectral | Yes | No | Any |
| **ACNN** | Any | Any | Yes | Spatial | Yes | Yes | Any |

Table 1: Comparison of different intrinsic learning models. Our ACNN model combines all the best properties of the other models. Note that OSD and ADD are local spectral descriptors operating with intrinsic geometric information of the shape and cannot be applied to arbitrary input, unlike the Random Forest (RF) and convolutional models.

ACNN, as any convolutional network, is applied in a point-wise manner on a function defined on the manifolds, producing a point-wise output that is interpreted as soft correspondence, as described below. Our intrinsic convolutional layer IC$Q$, with $Q$ output maps, is defined as follows and replaces the convolutional layer used in classical Euclidean CNNs with the construction (8). The IC$Q$ layer contains $PQ$ filters arranged in banks ($P$ filters in $Q$ banks); each bank corresponds to an output dimension. The filters are applied to the input as follows,

$$ f_q^{\text{out}}(x) = \sum_{p=1}^{P} (f_p^{\text{in}} * a_{qp})(x), \quad q = 1, \ldots, Q, \tag{9} $$

where $a_{qp}(\theta, t)$ are the learnable coefficients of the $p$th filter in the $q$th filter bank. A visualization of such filters is available in the supplementary material.

Overall, the ACNN architecture combining several layers of different type, acts as a non-linear parametric mapping of the form $\mathbf{f}_\Theta(x)$ at each point $x$ of the shape, where $\Theta$ denotes the set of all learnable parameters of the network. The choice of the parameters is done by an optimization process, minimizing a task-specific cost, and can thus be rather general. Here, we focus on learning shape correspondence.

**Learning correspondence** Finding correspondence in a collection of shapes can be cast as a labelling problem, where one tries to label each vertex of a given *query* shape $X$ with the index of a corresponding point on some *reference* shape $Y$ [20]. Let $n$ and $m$ denote the number of vertices in $X$ and $Y$, respectively. For a point $x$ on a query shape, the output of ACNN $\mathbf{f}_\Theta(x)$ is $m$-dimensional and is interpreted as a probability distribution ('soft correspondence') on $Y$. The output of the network at all the points of the query shape represents the probability of $x$ mapped to $y$.

Let us denote by $y^*(x)$ the ground-truth correspondence of $x$ on the reference shape. We assume to be provided with examples of points from shapes across the collection and their ground-truth correspondence, $\mathcal{T} = \{(x, y^*(x))\}$. The optimal parameters of the network are found by minimizing the *multinomial regression loss*

$$ \ell_{\text{reg}}(\Theta) \quad = \quad - \sum_{(x, y^*(x)) \in \mathcal{T}} \log \mathbf{f}_\Theta(x, y^*(x)). \tag{10} $$

## 5   Results

In this section, we evaluate the proposed ACNN method and compare it to state-of-the-art approaches. Anisotropic Laplacians were computed according to (5). Heat kernels were computed in the frequency domain using all the eigenpairs. In all experiments, we used $L = 16$ orientations and the anisotropy parameter $\alpha = 100$. Neural networks were implemented in Theano [2]. The ADAM [11] stochastic optimization algorithm was used with initial learning rate of $10^{-3}$, $\beta_1 = 0.9$, and $\beta_2 = 0.999$. As the input to the networks, we used the local SHOT descriptor [21] with $544$ dimensions and using default parameters. For all experiments, training was done by minimizing the loss (10). For shapes with 6.9K vertices, Laplacian computation and eigendecomposition took $1$ sec and $4$ seconds per angle, respectively on a desktop workstation with 64Gb of RAM and i7-4820K CPU. Forward propagation of the trained model takes approximately $0.5$ sec to produce the dense soft correspondence for all the vertices.

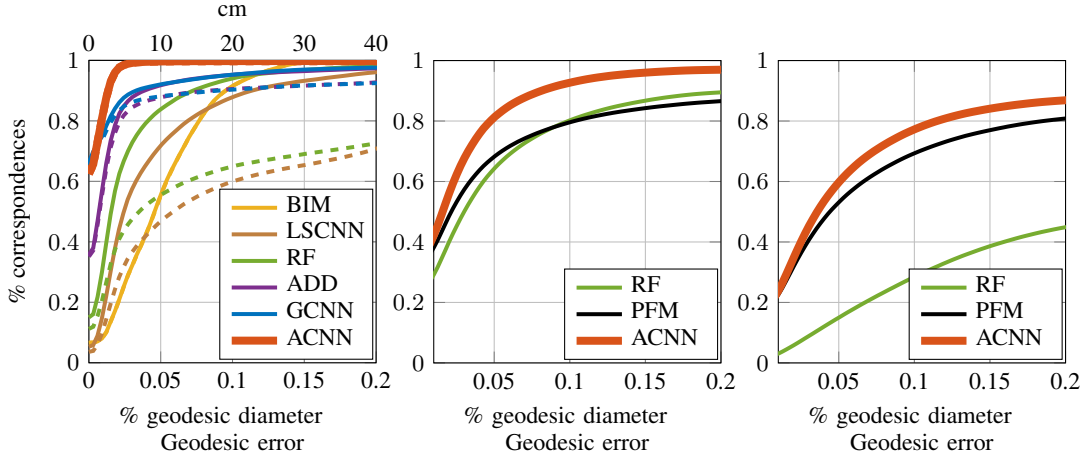

Figure 2: Performance of different correspondence methods, left to right: FAUST meshes, SHREC'16 Partial cuts and holes. Evaluation of the correspondence was done using the Princeton protocol.

**Full mesh correspondence** We used the FAUST humans dataset [3], containing 100 meshes of 10 scanned subjects, each in 10 different poses. The shapes in the collection manifest strong non-isometric deformations. Vertex-wise groundtruth correspondence is known between all the shapes. The zeroth FAUST shape containing 6890 vertices was used as reference; for each point on the query shape, the output of the network represents the soft correspondence as a 6890-dimensional vector which was then converted to point correspondence with the technique explained in Section 4. First 80 shapes for training and the remaining 20 for testing, following verbatim the settings of [16]. Batch normalization [9] allowed to effectively train larger and deeper networks. For this experiment, we adopted the following architecture inspired by GCNN [16]: FC64+IC64+IC128+IC256+FC1024+FC512+Softmax. The soft correspondences produced by the net were refined using functional map [18]. We refer to the supplementary material for the details. We compare to Random Forests (RF) [20], Blended Intrinsic Maps (BIM) [10], Localized Spectral CNN (LSCNN) [4], and Anisotropic Diffusion Descriptors (ADD) [5].

Figure 2 (left) shows the performance of different methods. The performance was evaluated using the Princeton protocol [10], plotting the percentage of matches that are at most $r$-geodesically distant from the groundtruth correspondence on the reference shape. Two versions of the protocol consider intrinsically symmetric matches as correct (symmetric setting, solid curves) or wrong (asymmetric, more challenging setting, dashed curves). Some methods based on intrinsic structures (e.g. LSCNN or RF applied on WKS descriptors) are invariant under intrinsic symmetries and thus cannot distinguish between symmetric points. The proposed ACNN method clearly outperforms all the compared approaches and also perfectly distinguishes symmetric points. Figure 3 shows the pointwise geodesic error of different correspondence methods (distance of the correspondence at a point from the groundtruth). ACNN shows dramatically smaller distortions compared to other methods. Over 60% of matches are exact (zero geodesic error), while only a few points have geodesic error larger than 10% of the geodesic diameter of the shape [1]. Please refer to the supplementary material for an additional visualization of the quality of the correspondences obtained with ACNN in terms of texture transfer.

**Partial correspondence** We used the recent very challenging SHREC'16 Partial Correspondence benchmark [7], consisting of nearly-isometrically deformed shapes from eight classes, with different parts removed. Two types of partiality in the benchmark are *cuts* (removal of a few large parts) and *holes* (removal of many small parts). In each class, the vertex-wise groundtruth correspondence between the full shape and its partial versions is given. The dataset was split into training and testing disjoint sets. For cuts, training was done on 15 shapes per class; for holes, training was done on 10 shapes per class. We used the following ACNN architecture: IC32+FC1024+DO(0.5)+FC2048+DO(0.5)+Softmax. The soft correspondences produced by the net were refined using partial functional correspondence [19]. We refer to the supplementary mate-

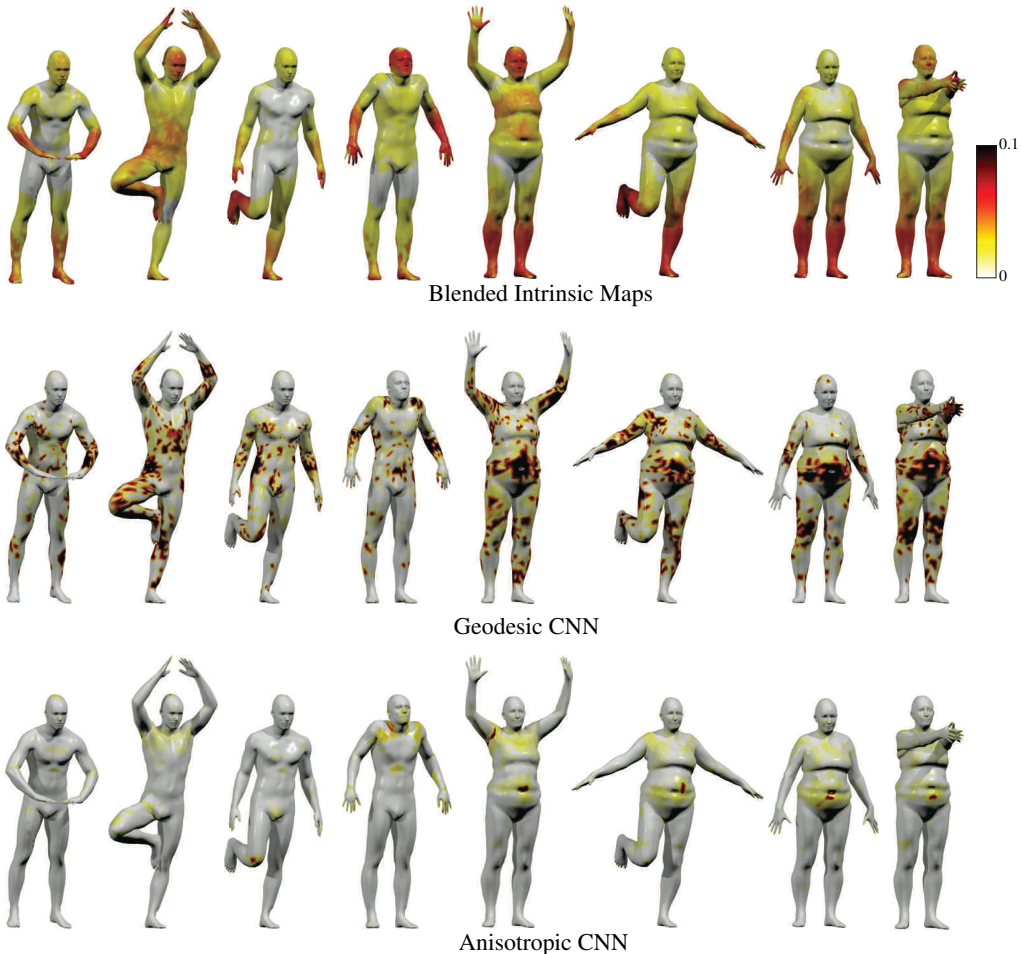

Figure 3: Pointwise geodesic error (in % of geodesic diameter) of different correspondence methods (top to bottom: Blended Intrinsic Maps, GCNN, ACNN) on the FAUST dataset. Error values are saturated at 10% of the geodesic diameter. Hot colors correspond to large errors.

rial for the details. The dropout regularization, with $\pi_{\mathrm{drop}} = 0.5$, was crucial to avoid overfitting on such a small training set. We compared ACNN to RF [20] and Partial Functional Maps (PFM) [19]. For the evaluation, we used the protocol of [7], which closely follows the Princeton benchmark.

Figure 2 (middle) compares the performance of different partial matching methods on the SHREC'16 Partial (cuts) dataset. ACNN outperforms other approaches with a significant margin. Figure 4 (top) shows examples of partial correspondence on the horse shape as well as the point-wise geodesic error. We observe that the proposed approach produces high-quality correspondences even in such a challenging setting. Figure 2 (right) compares the performance of different partial matching methods on the SHREC'16 Partial (holes) dataset. In this setting as well, ACNN out-performs other approaches with a significant margin. Figure 4 (bottom) shows examples of partial correspondence on the dog shape as well as the pointwise geodesic error.

## 6   Conclusions

We presented Anisotropic CNN, a new framework generalizing convolutional neural networks to non-Euclidean domains, allowing to perform deep learning on geometric data. Our work follows the very recent trend in bringing machine learning methods to computer graphics and geometry processing applications, and is currently the most generic intrinsic CNN model. Our experiments show that ACNN outperforms previously proposed intrinsic CNN models, as well as additional state-of-the-art methods in the shape correspondence application in challenging settings. Being a generic model, ACNN can be used for many other applications. The most promising future work direction is applying ACNN to learning on graphs.

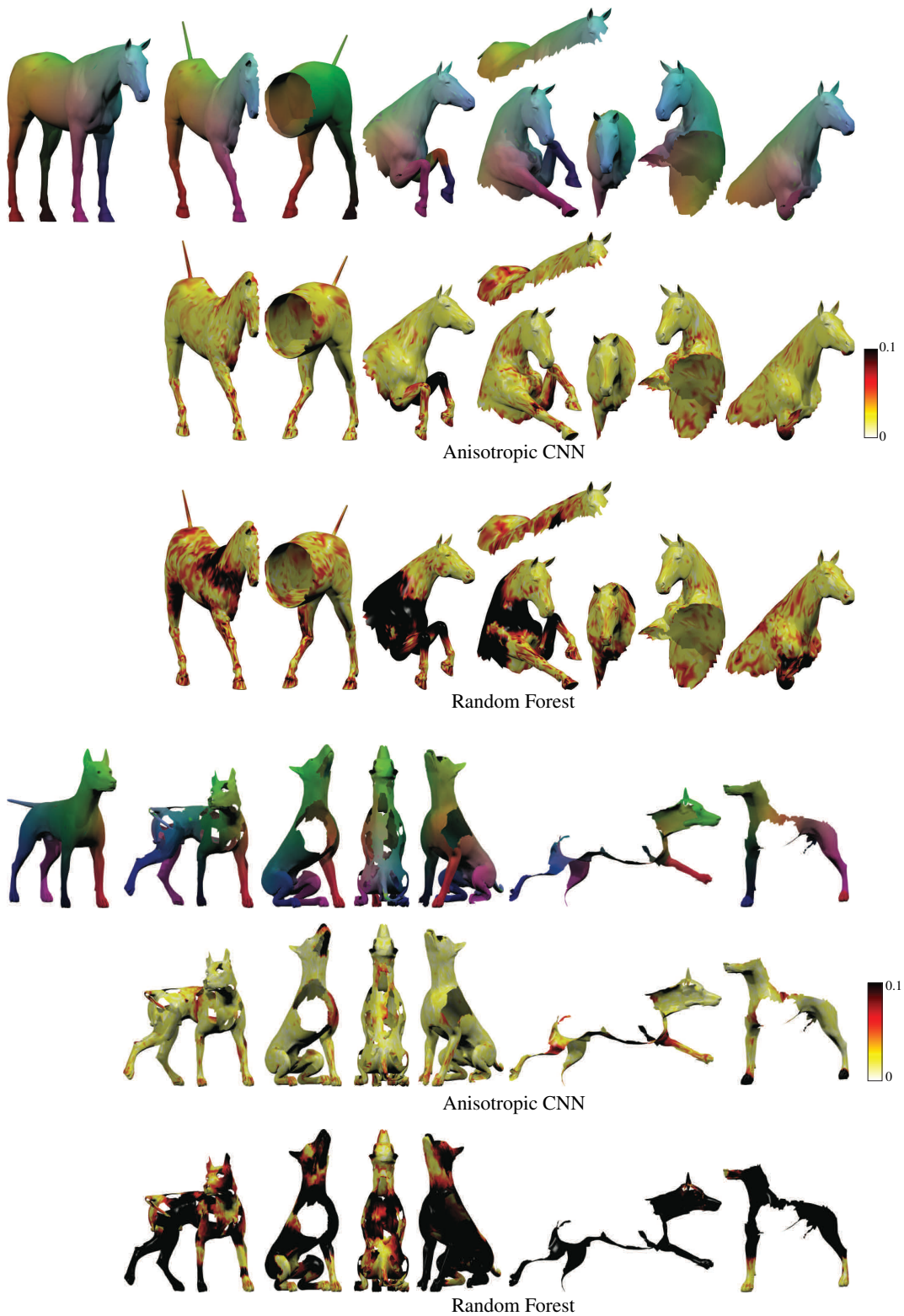

Figure 4: Examples of partial correspondence on the SHREC'16 Partial cuts (top) and holes (bottom) datasets. Rows 1 and 4: correspondence produced by ACNN. Corresponding points are shown in similar color. Reference shape is shown on the left. Rows 2, 5 and 3, 6: pointwise geodesic error (in % of geodesic diameter) of the ACNN and RF correspondence, respectively. Error values are saturated at 10% of the geodesic diameter. Hot colors correspond to large errors.

## Acknowledgments

The authors wish to thank Matteo Sala for the textured models. This research was supported by the ERC Starting Grant No. 307047 (COMET), a Google Faculty Research Award, and Nvidia equipment grant.

## Footnotes

[1]Per subject leave-one-out produces comparable results with mean accuracy of $59.6 \pm 3.7\%$.

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
