[Supplementary Material]

# Learning shape correspondence with anisotropic convolutional neural networks supplementary material

## 1 Correspondence refinement

**Full correspondence.** The most straightforward way to convert the soft correspondence $f(x, y)$ produced by ACNN into a point-wise correspondence is by assigning $x$ to

$$\hat{y}(x) = \arg\max_{y \in Y} f(x, y). \tag{1}$$

We instead use a slightly more elaborate scheme to refine the soft correspondences produced by ACNN. First, we select a subset of points $I = \{x : c(x) > \tau_{\mathrm{th}}\}$, where $c(x) = \max_{y \in Y} f(x, y) \in [0, 1]$, at which the confidence of the predicted correspondence exceeds some threshold $\tau_{\mathrm{th}}$. Second, we use this subset of corresponding points to find a functional map [1] between $L^2(X)$ and $L^2(Y)$ by solving the linear system of $|I|k$ equations in $k^2$ variables,

$$\boldsymbol{\Phi}_I \mathbf{C} = \boldsymbol{\Psi}_I, \tag{2}$$

where $\boldsymbol{\Phi}_I$ and $\boldsymbol{\Psi}_I$ are the first $k$ Laplace-Beltrami eigenfunctions of shapes $X$ and $Y$, respectively, sampled at the subset of corresponding points (represented as $|I| \times k$ matrices). Third, given the functional map $\mathbf{C}^*$, we produce a new point-wise correspondence by matching $\boldsymbol{\Phi}\mathbf{C}^*$ and $\boldsymbol{\Psi}$ in the $k$-dimensional eigenspace, $y(x) = \arg\max_{y \in Y} \|(\phi_1(x), \dots, \phi_k(x))\mathbf{C}^* - (\psi_1(y), \dots, \psi_k(y)\|_2$.

**Partial correspondence.** A similar procedure is employed in the setting of partial correspondence, where instead of the computation of a functional map, we use the recently introduced *partial functional map* [2].

## 2 ACNN filter visualization

In Figure 1 we visualize the filter learned by ACNN. It is interesting to notice the directional information captured by the filters.

Figure 1: Examples of filters in the first IC layer learned by the ACNN (hot and cold colors represent positive and negative values, respectively).

## 3 ACNN correspondence visualization

In Figure 2 we show the quality of the correspondences obtained with ACNN in terms of texture transferring. The result is quite remarkable: not only the corresponding regions are mapped correctly, but the single vertices inside those regions are consistent (the letters on the target shapes have little to null distortion).

Figure 2: Examples of correspondence on the FAUST humans dataset obtained by the proposed ACNN method. Shown is the texture transferred from the leftmost reference shape to different subjects in different poses by means of our correspondence. The correspondence is nearly perfect (only very few minor artifacts are noticeable).