[Reviews · NeurIPS 2016]

Reviewer 1

Summary

The authors propose the "anisotropic neural network", a novel method for intrinsic deep learning, i.e. directly computing functions computed on a non-Euclidean domain. They are particularly interested in computing correspondences between different non-Euclidean objects, e.g. computing corresponding points on a body in different poses. I'm not familiar with this area, but I think that the anisotropic network works as follows: You learn (?) several heat kernels, which each implement a mapping from the intrinsic space onto a polar system of coordinates. For each of the heat kernels, several filters are applied. The i'th filter from each kernel is summed to form the i'th output plane. The authors show qualitative results on several benchmarks, which strongly outperform the prior GCNN approach.

Qualitative Assessment

The paper is presented very clearly, giving ample references, excellent diagrams, and clear explanation of the problem, theoretical work, and evaluation. I wonder if some of the explanation in Sec. 2 and 3 can be made a bit more accessible to a wider audience. I don't have suitable background to evaluate the theoretical approach or comparison with other methods for instrinsic learning on non-Euclidean domains. The qualitative results, both in isolation and in comparison to prior work, look quite remarkable to me. The correspondence mapping shown in the supplementary material is particularly impressive. Even though I am not familiar with this area, I would love to learn more about this work.

Confidence in this Review

1-Less confident (might not have understood significant parts)


Reviewer 2

Summary

This paper introduces Anisotropic CNN, a new framework generalizing convolutional neural networks to non-Euclidean domains, allowing to perform deep learning on geometric data. This is basically a machine learning application of computer graphics and geometry processing applications. The experiments (on benchmarks such as SHREC’16 Partial) show that ACNN outperforms previously proposed intrinsic CNN models, as well as additional state-of-the-art methods in the shape correspondence application in challenging settings.

Qualitative Assessment

This paper is generally well written. The application of finding shape correspondence is quite important for many fields, especially for shape analysis related fields. The proposed approach seems quite effective as well. The experiments are convincing in general, although some additional improvements can be made. For example, more datasets and also comparisons against more methods. It is also interesting to try more advanced network architectures and see if this can improve the performance. It is mentioned that "First 80 shapes for training and the remaining 20 for testing". I assume these 80 shapes are from 8 subjects, and the 20 testing ones are from the remaining 2 subjects? In this case, it makes more sense to use leave-one-out considering that there are only 10 subjects. Overall the quality of this paper is good and I didn't notice any major flaw. My main concern is that whether this fits the scope of NIPS. This is more like a graphics paper (most compared methods are published in that community as well). Other than this, I have no concerns.

Confidence in this Review

1-Less confident (might not have understood significant parts)


Reviewer 3

Summary

The paper aims at learning intrinsic correspondence using convolutional neural network, The proposed method, Anisotropic Convolutional Neural Network (ACNN), is a variant of CNN that can deal with non-Euclidean domains. Overall, the paper is well written.

Qualitative Assessment

(1) The novelty. The paper is not so novel. Basically, the proposed method extends ADD [5] to deep learning framework, which is proposed in GCNN [16]. Meanwhile, this paper is not the first one to learn shape correspondence using deep learning. In this sense, the importance of this paper might be limited. (2) The experiments. The experimental comparison is sufficient. However, only one method [5] is based on deep learning as the proposed method. Such a comparison is somewhat unfair. As the most relevant paper, why [16] is not compared?

Confidence in this Review

2-Confident (read it all; understood it all reasonably well)


Reviewer 4

Summary

This paper proposes a novel type of CNN that operates on Riemannian manifolds. Different from existing such manifold CNNs, the proposed method uses anisotropic diffusion kernels to reparameterize functions defined in a local neighborhood of a point on surface. This new architecture, called ACNN, is tested on the point correspondence problem between shapes. Experiments show that it outperforms previous approaches on standard benchmark datasets of both complete and partial 3D shapes.

Qualitative Assessment

It is still an open problem of what the best practice is to design convolutional neural networks on manifold. Unlike CNNs on Euclidean space, parallel translation and the statistical invariance coupled with this operation are not well defined. This core problem has stimulated several recent attempts, and this paper also falls into this category. In my opinion, it is reasonable to use heat diffusion process to define the proximity of a point. However, it is still unclear why the invariance assumption should hold using this parameterization. I would be more satisfied if the authors could do more in-depth statistical analysis over the distribution of the output from the patch operator, as defined in Eq (7).

Confidence in this Review

3-Expert (read the paper in detail, know the area, quite certain of my opinion)


Reviewer 5

Summary

This paper proposes a novel model called Anisotropic Convolutional Neural Network (ACNN), which generalizes classical convolutional neural networks to non-Euclidean domains. This work builds on two methods, namely, the Geodesic Convolutional Neural Network ("Geodesic Convolutional Neural Networks on Riemannian Manifolds") and the concept of Anisotropic Diffusion Descriptors. While sharing many foundational ideas, it describes a new "fused" approach resembling the idea of the Geodesic Convolutional Neural Network, but using kernels of the anisotropic Laplacian operator and avoiding a construction of geodesic local patches (which, if I understand everything correctly, appears to be an artificial and inconvenient construct). The suggested approach is demonstrated to outperform other approaches when applied to the problems of full and partial mesh correspondence.

Qualitative Assessment

> Technical quality. I believe the paper to be of a high technical quality. The calculations are sound and appear to not have any major flaws or mistakes (other than small inaccuracies outlined below). The experimental methods used to evaluate the proposed method seem to be appropriate. > Novelty/originality. This work builds on the publication "Geodesic Convolutional Neural Networks on Riemannian Manifolds" and the idea of Anisotropic Diffusion Descriptors. It shares the foundational concepts and notation with these articles, but proposes a novel "fused" approach combining the strengths of these two prior publications. While not entirely novel in its details, the combination of two approaches seems to be a successful idea and as such a promising "step in the right direction". > Potential impact or usefulness. This work proposes a model which shows a significantly improved performance compared to prior approaches and seems to show a potential of having a large impact in its specialized sub-field. > Clarity and presentation. The paper is well written. However, the clarity of the presentation is lacking, seemingly due to the fact that the authors expect the reader to be familiar with prior publications on Geodesic Convolutional Neural Networks and Anisotropic Diffusion Descriptors. Some of the minor inaccuracies/problems include: (a) the conductivity tensor defined in Eq. (2) should have an additional x-dependent multiplier; (b) in Eq. (7), D should have subscripts; (c) the subsection about the discretization of the anisotropic Laplacian operator should include at least some references (Anisotropic Diffusion Descriptors, ...?); (d) instead of "the solution of heat equation (1) at time t is obtained by applying the anisotropic heat operator ...", the sentence should read "the solution of heat equation (1) with D=D_{\alpha \theta} at time t is obtained by applying the anisotropic heat operator ..."; (e) although the meaning of the notation < a,b >_H should be clear for the majority of the readers, including a definition or a reference would improve the clarity of the article.

Confidence in this Review

1-Less confident (might not have understood significant parts)


Reviewer 6

Summary

This paper uses anisotropic heat kernel as local intrinsic filters to construct convolutional neural network. The network is used to construct correspondence between deformed shapes. Experiments on FAUST and SHREC'16 show good performance for both full shape and partial shape correspondence.

Qualitative Assessment

Compared to GCNN, this paper simplifies the construction of discrete patch operator. It also avoided using angular max pooling by using local curvature direction as reference frame. However the paper did not make it clear why ACNN does not need mesh as input. In construction of discrete anisotropic Laplacian a mesh is used with its faces and edges.

Confidence in this Review

2-Confident (read it all; understood it all reasonably well)